# Associations between Cardiovascular Risk Factors and Timed Up and Go Test for Elderly Participants in Public Physical Activity Programs

**DOI:** 10.3390/ijerph21080993

**Published:** 2024-07-29

**Authors:** Marília Salete Tavares, Sara Lucia Silveira de Menezes, Emanuel Davi Farias Ribeiro, Marco Orsini, Fábio Augusto d’Alegria Tuza, Paulo Henrique de Moura, Dinah Vasconcelos Terra, Adalgiza Mafra Moreno

**Affiliations:** 1Exercise Physiology Laboratory, Postgraduate Program in Physical Activity Sciences, Universidade Salgado de Oliveira, Niterói 24030-060, Brazil; saraposgraduacao@yahoo.com.br (S.L.S.d.M.); emanueldribeiro97@gmail.com (E.D.F.R.); 2Health and Aging Research Group, Universidade Iguaçu, Nova Iguaçu 26275-580, Brazil; orsinimarco@hotmail.com (M.O.); fabiotuza@gmail.com (F.A.d.T.); paulohdemoura@gmail.com (P.H.d.M.); 3Health Sciences Center, Universidade Federal do Rio de Janeiro, Rio de Janeiro 21941-901, Brazil; 4Education Department, Universidade Federal Fluminense, Niterói 24220-900, Brazil; dvterra@id.uff.br

**Keywords:** cardiovascular risk factors, elderly, anthropometric indicators, Timed Up and Go test

## Abstract

Variables such as body mass index (BMI), waist circumference (WC), and waist/height ratio (WHtR) are used to assess cardiovascular risks associated with abdominal obesity. The Timed Up and Go (TTUG) test assesses mobility and the risk of falls, especially in the elderly and individuals with physical limitations. The objective was to correlate anthropometric indicators of cardiovascular risk (BMI, WC, WHtR) with performance on the TTUG test in elderly people who practice physical activity. Methods: Observational, cross-sectional study, CAAE: 27116319.1.0000.8044. Active elderly participants in a public physical exercise program “Project 60 Up”, promoted by the Municipal Secretariat for the Elderly of the City of Niterói, RJ, Brazil, were evaluated. Elderly people of both sexes, with independent locomotion and preserved cognitive status, were included and those with visual, hearing, severe mobility difficulties or neurological sequelae and imbalances were excluded. Anthropometric data were collected: BMI, WC, WHtR, and the TTUG test was performed. Results: In the sample composed of 55 elderly people of both sexes, with an average age of 68 years, the means of the variables were: body weight (67.8 ± 12.7 kg), height (157.2 ± 8.4 cm), TTUG (10.3 ± 2 s), WC (93.3 ± 10.9 cm), WHtR (0.59), and BMI (27.4 ± 4.4). The correlations were between TTUG and age (r = −0.24, *p* = 0.69), TTUG and BMI (r = 0.111, *p* = 0.426), and WC and WHtR (r = 0.885, *p* < 0.000). Weak correlations were observed between TTUG and BMI and WC and WHtR, indicating that factors other than BMI and abdominal fat accumulation may have a greater influence on performance on the TTUG test. In this specific context of elderly people participating in a physical activity program, the results found may have been shaped by the very nature of the interventions carried out in the program, with factors such as muscular strength, flexibility and balance preserved and acquired through the practice of regular physical exercise. Despite the benefits of physical activity, managing weight and abdominal fat remains challenging for elderly individuals with high anthropometric measurements. Conclusions: Although many elderly people have anthropometric measurements above average levels, the results indicate that no negative influence on their performance on the TTUG was observed. However, the limitation of the sample size and the underrepresentation of elderly people ≥ 80 years and of men highlight the need for future studies with larger and more balanced samples to confirm these results.

## 1. Introduction

The aging process is intrinsic to human life and encompasses transformations over time [1]. This phase is marked by the progressive accumulation of molecular and cellular damage, resulting in significant health-related challenges, including an increased risk of cardiovascular disease and falls [1,2]. As an individual ages, the musculoskeletal system undergoes physiological changes that compromise its functionality, such as the loss of muscle mass and the increase in adipose tissue, which are common in the aging process [2,3,4].

These changes result in a decline in physical abilities, including a reduction in muscle strength, flexibility, coordination, and joint mobility [4,5]. These factors lead to an increase in postural instability and have a substantial impact on mobility and autonomy, thereby increasing the risk of falls among the elderly [6,7].

Furthermore, cardiovascular health can also be compromised due to the stiffening of the arteries, increased blood pressure, and a decrease in maximum cardiac output, closely linked to a reduction in maximum heart rate and other age-related changes [8,9]. These factors contribute to a greater risk of cardiovascular disease, which in turn can negatively affect the body’s ability to maintain balance and respond adequately to postural challenges, thus increasing the risk of falls [5,8,9,10]. Abdominal adiposity is associated with musculoskeletal dysfunction and reduced mobility, both of which contribute to conditions such as sarcopenia (the loss of muscle mass) and increased lordosis, which directly affect the ability to maintain balance and perform safe movements [4].

Several studies have shown an association between metabolic diseases and abdominal obesity, and its high occurrence worldwide has been confirmed [11,12]. Abdominal obesity is also associated with an increased risk of cardiovascular diseases such as hypertension, which can affect vascular and neurological health and contribute to falls [11,12]. Anthropometric indicators such as the body mass index (BMI), waist circumference (WC), and waist-to-height ratio (WHtR) are important for assessing abdominal adiposity, which is strongly linked to risk conditions for hypertension and diabetes [11,12,13,14,15]. Contrariwise, the Timed Up and Go (TTUG) test is widely recognized as a very useful tool for assessing mobility and the risk of falls [16]. This test includes a series of simple activities that reflect the risk of falls, the fear of falling, overall functional capacity, and the motor coordination needed to perform everyday tasks safely [16,17].

The elderly population faces a high risk of both cardiovascular diseases and falls, both of which are influenced by the physiological changes associated with aging, such as the loss of muscle mass and the gain of adipose tissue [4,15]. Physical exercise can be beneficial in both cases, improving balance and maintaining muscle mass and strength, which are important for stability and preventing falls, as well as improving blood circulation and lowering blood pressure, which reduces the risk of cardiovascular disease [1,18,19].

Understanding aging as a multifaceted and highly individualized process, with the early identification of health risk factors and the adoption of preventive strategies and personalized interventions, can help minimize the negative impacts of this process and promote healthier and more active aging [1,18]. The implementation of specific physical exercise programs for the elderly, such as those offered in community and health centers, can have a significant positive impact on the prevention of falls and cardiovascular risks. Therefore, promoting and facilitating access to regular physical activity should be a priority in public health policies aimed at the elderly population [18].

In this context, Niterói, a city located in the state of Rio de Janeiro, is known for its high quality of life [20]. The city is home to several universities and educational centers, resulting in a high level of education, especially among the elderly, whose literacy rate is high compared to other regions of Brazil [20,21,22]. The city of Niterói has the highest percentage of elderly people in the state, offering a good health and leisure infrastructure, with community centers and public parks where the elderly population can socialize and practice physical activities, keeping active throughout their lives [20,21].

Public programs that offer physical activity to the elderly are becoming increasingly popular, seeking to promote active aging, reduce the risk of falls, and improve the health and well-being of this population [21]. Although numerous studies have been carried out on the benefits of government physical activity programs, many of them have been qualitative in nature [18]. It is therefore necessary to carry out research with quantitative data to complement existing knowledge.

Based on these considerations, the hypothesis was developed that the TTUG test is reliable for assessing the decline in physical mobility and risk of falls over time and correlating it with anthropometric indicators of cardiovascular risk in elderly people who practice physical activities. Therefore, this study is justified by the need to better understand the complex interactions between cardiovascular risk indicators and performance on the TTUG test in an active elderly population, providing data that can be used to develop more effective public health interventions tailored to the specific needs of the elderly in public physical activity programs.

The aim of this study was therefore to investigate possible associations between anthropometric indicators of cardiovascular risk and performance in the Timed Up and Go test in elderly participants in a public physical activity program in the city of Niterói.

## 2. Materials and Methods

### 2.1. Design Study

This was an observational, cross-sectional, and analytical study, approved by the Ethics Committee, CAAE: 67496423.6.0000.8044. The sample consisted of 55 elderly people of both sexes, aged 60 or over, who practiced physical activities. All the elderly included in the study were given a verbal explanation of the study, its risks and benefits, and agreed to take part in the study by signing the Free and Informed Consent Form (FICF). The evaluations took place in six different centers of the 60 Up project, a physical activity program for the elderly, promoted by the Municipal Department for the Elderly of the Niterói City Hall, RJ, Brazil.

### 2.2. Description of Exercises and Intensities

The exercise program includes various activities organized into different phases with variations in intensity. The initial warm-up involves five minutes of deep breathing, dynamic stretching, and walking in place, all of low to moderate intensity. In the main part, squats with poles and deadlifts are performed, each for five minutes, at moderate to high intensity. Following a ten-minute break, there are rhythmic exercises to music, simple dance steps, and choreographed sequences, each for five minutes, at moderate to high intensity. The end of the activities includes static stretching and deep, low-intensity breathing. This program is designed to gradually increase heart rate and breathing, offering a complete and balanced workout for the elderly [21].

-Low: Exercises that require little physical effort, ideal for starting and finishing activities;-Moderate: Activities that increase heart rate and breathing, but still allow conversation;-Moderate to High: Exercises that significantly increase heart rate and breathing, making conversation difficult during the activity.

### 2.3. Sample Selection

This was a non-probabilistic, convenience sample. Eligible for this study were active elderly people participating in a public physical exercise program, held 5 days a week (Monday to Friday), lasting 60 min per session.

### 2.4. Inclusion and Exclusion Criteria

Inclusion criteria were as follows: participants in the exercise program for more than 30 consecutive days, in the proposed age group, and meeting specific inclusion criteria. These criteria included having independent locomotion, preserved cognitive status, and the ability to understand and follow instructions related to performing the Timed Up and Go (TTUG) test.

The exclusion criteria involved elderly people with functional disabilities, such as those requiring a wheelchair or walking aids, individuals with severe visual or hearing difficulties, and neurological sequelae and imbalances. Participants with flu-like symptoms were also excluded to avoid the possibility of contagion among the other participants. In addition, flu-like symptoms such as fever, fatigue, muscle pain, and breathing difficulties can compromise the safety and well-being of participants and significantly affect physical performance and the ability to perform the TTUG test proposed in the study, negatively influencing the results of the assessments [23].

### 2.5. Instruments Used to Assess Anthropometric Measurements

Anthropometric measurements were taken (weight, height, and waist circumference). A digital scale, stadiometer, and measuring tape graduated in centimeters were used. The volunteers’ height was measured to an accuracy of 0.1 cm using a compact, portable Sanny tape measure fixed to the wall, with a measuring range of 0 cm to 2.10 m. Before starting the measurement, the volunteer was asked to remove their shoes to avoid altering the measurement. For the measurement, the participant stood with their heels together and their legs parallel. The arms were kept relaxed at the side of the body, with the palms of the hands facing inwards. In addition, the evaluated volunteer kept their head in a plane where the lowest point of the orbital margin was aligned horizontally with the highest point of the margin of the external acoustic meatus, known as the Frankfurt plane [24]. Once this was completed, the cursor was moved to the head, exerting enough pressure to compress the hair and the measurement was recorded manually.

Body weight was assessed by electrical bioimpedance using the ITeknic IK-PCA001 (ITEKNIC Shenzhen NerabyExpress, Shenzhen, China) intelligent body scale. Participants were instructed to keep a 3 h fast, to do the assessment with an empty bladder, and not to do any physical activity in the last 12 h. The participants were assessed while in an orthostatic position, with their arms extended at their sides and their feet apart, barefoot and parallel, wearing light clothing and no adornments. The collected height and weight information was used to calculate BMI, determined using the formula BMI = Weight/Height^2^ [12]. 

Although widely used, BMI does not differentiate between lean mass and body fat, which limits its ability to assess abdominal adiposity, which is a critical risk factor in cardiovascular health [12,13,14,25]. According to the World Health Organization (WHO), individuals with a BMI < 18.5 kg/m^2^ are classified as underweight; eutrophy—BMI between 18.5 kg/m^2^ and 24.9 kg/m^2^; overweight—BMI of 25 kg/m^2^; and those with a BMI above 29.9 kg/m^2^ were classified as obese [13,14,25,26,27].

Waist circumference was measured using a non-elastic tape measure positioned parallel to the floor, placed at the midpoint between the tenth rib and the iliac WHtR. During the measurement, the volunteers were instructed to exhale completely to ensure that the reading was accurate [25,26]. Cut-off points were considered according to the degree of risk for cardiovascular disease: increased risk for women (WC > 80 cm) and men (WC > 94 cm), and very increased risk for women (WC > 88 cm) and men (WC > 102 cm) [11,25,26,27].

Height and waist circumference information was used to establish the waist-to-height ratio using the formula: WHtR = Height(cm)/WC(cm) [23]. A recommended WHtR value to reduce the risk of metabolic diseases should be less than 0.5. However, the limits may vary according to age and other individual characteristics [11,25,26,27].

### 2.6. Performing the Timed Up and Go Test

The TTUG test assesses the time required for an individual to perform a sequence of actions, which includes getting up from a standard-height chair, walking a distance of 3 m, turning around, returning to the chair, and sitting down again. The measured time, in seconds, serves as a practical measure for assessing functional mobility, the risk of falls, and dynamic balance in the elderly and adults with motor limitations [16]. A time of less than 10 s is considered normal; between 10 and 19 s indicates a moderate risk of falls, and above 19 s is considered a high risk of falls [16,17].

The TTUG test was carried out in a covered, flat, and well-lit corridor, following the guidelines of the original study [27]. To carry out the TTUG test, a tape measure was fixed to the floor, marking a distance of 3 m from the starting point (a chair).

At the end of the 3 m, a cone was placed to serve as a reference point around which the participants had to go before returning to the starting point [16,28]. Prior to the test, the researcher explained each step of the procedure in detail to the participants. To ensure complete understanding, the researcher demonstrated the test, showing how to get up from the chair, walk to the cone, go around it, and return to the chair.

Each elderly person began the test sitting on a chair, with their back against the backrest and their feet touching the floor. The test was started at the verbal command “Go” given by the assessor, and the chronometer was activated manually. The participant rose from the chair, walked the pre-defined distance of 3 m, walked around the cone, returned, and sat down on the chair. The evaluator manually recorded the total time elapsed. The test was repeated if the participant was unable to complete the task correctly on the first attempt due to distractions, interruptions, or other factors unrelated to physical ability.

### 2.7. Statistical Analysis

Statistical data were organized and tabulated in Microsoft Excel^®^365 MSO spreadsheets (Version 2406) and analyzed using descriptive and exploratory statistics, presented as means, standard deviation, as well as analysis of absolute and relative frequencies for categorical variables. As part of the statistical analysis, the Shapiro–Wilk normality test was used to assess the distribution of the data.

Additionally, Pearson’s correlation was applied to investigate correlations between anthropometric indicators and the results obtained in the TTUG test, establishing a significance level of α = 0.05.

## 3. Results

The study took place between March and July 2023, a sample consisting of 55 elderly individuals, with an average age of 68 years, of both sexes, participating in the “60 Up” physical activity program, was evaluated. Among this population, a significant proportion (87%) were women, with an average age of 68 years. The men evaluated had an average age of 74 years.

To provide a detailed analysis of the data, Table 1 below shows the distribution of participants divided into three age groups: from 60 to 69 years old, from 70 to 79 years old, and from 80 years old and above, with the evaluation results expressed as mean and standard deviation.

There was a predominance of individuals classified as young elderly, covering the 60–69 age group, representing 52% of the total. There was a moderate negative correlation between BMI and age (−0.252), indicating that, in the context of this specific population, older people tend to have slightly lower body weights (Figure 1).

## 4. Discussion

The aim of this study was to correlate anthropometric indicators with performance in the TTUG test in elderly people taking part in a public program of regular physical activity. Obesity and the associated anthropometric variables, classified as cardiovascular risk factors, affect various age groups, especially the elderly. In this population, the incidence of abdominal obesity contributes to an increase in cardiac events, falls, and low functionality [8,9,10].

The results showed stronger positive correlations between waist circumference and body mass index (r = 0.705), waist circumference and waist-to-height ratio (r = 0.885), and body mass index and waist-to-height ratio (r = 0.761). However, the correlations between the TTUG test and these same anthropometric indicators (BMI, WC, and WHtR) were weak in this sample. This highlights the low influence of abdominal fat on the mobility of the elderly people assessed, who are regular exercisers.

According to a study by Brech et al. (2021) [29], lean mass reduces postural oscillations. On the other hand, fat mass negatively affects the dynamic postural balance of elderly women, showing that the greater the accumulation of body fat, the worse the balance. This finding reinforces the importance of maintaining lean body mass and reducing body fat in order to improve functionality and prevent falls in the elderly [30].

To conduct a detailed data analysis, as can be seen in Table 1, the distribution of the participants was divided into three age groups: 60 to 69 years old, 70 to 79 years old, and ≥80 years old, with the results of the evaluations expressed as mean and standard deviation. This division made it possible to identify possible trends and differences between the age groups regarding the results of anthropometric measurements and the TTUG test, providing a deeper understanding of how aging affects anthropometric characteristics and mobility.

The elderly people evaluated generally showed good results in the TTUG test, with the 60-to-69-year-old group scoring below 10 s (9.7 s), indicating good mobility and a low risk of falls, which is to be expected given that individuals in this age group generally have less muscle loss and fewer mobility problems compared to older groups. However, the BMI (28.1 kg/m^2^), WC (94.6 cm), and WHtR (0.6 cm) results were above the appropriate averages, suggesting that many of these individuals are overweight, which indicates the need for interventions to control weight and prevent future health problems.

The group of elderly people aged between 70 and 79 represents a transitional stage, where the effects of aging begin to intensify. They had a slightly longer average time in the TTUG test than the 60–69 age group, reflecting a slight decline in functional mobility. Anthropometric measurements, such as BMI (27.1 kg/m^2^) and WC (93.2 cm), were also indicative of overweight, which can influence the general health and mobility of these individuals. However, the WHtR result (0.5 cm) was within the parameters considered adequate.

The group aged ≥80 years had the longest average TTUG time, at 11.2 s, indicating a moderate risk of falls and lower functional mobility. In addition, the data showed a lower BMI (21.1 kg/m^2^), a lower WC (81.5 cm), and the ideal WHtR result (0.5 cm). The lower body mass of this group, compared to the other younger elderly groups, possibly reflects the loss of muscle mass and weight common in this age group, which can be associated with frailty and a greater risk of falls.

The ELSA-Brazil study examined parameters such as waist circumference, waist-to-hip ratio, conicity index, lipid accumulation product, and visceral adiposity index. Notably, these data are close to the measurement of visceral fat obtained through imaging methods, establishing an association between greater mid-intimal thickness and greater prevalence of abdominal fat [30]. These results highlight the importance of regular physical activity programs for the elderly, not only to improve functional mobility, but also to mitigate the negative impacts of abdominal obesity on cardiovascular health and physical function in the elderly population [21].

According to a study by Zeng Q et al. (2014), the ideal cut-off values for BMI are: 24.0 and 23.0 kg/m^2^; 85.0 and 75.0 cm for WC; and between 0.50 and 0.48 for WHtR, for men and women, respectively. However, as can be seen in Table 1, the results of the volunteers assessed in this study were higher than those presented in the literature [27]. Despite these elevated anthropometric values, it is important to note that regular physical activity may have had a significant positive impact on the body composition, mobility, and cardiovascular health of the elderly people evaluated [18,19].

According to Bohannon’s study (2006), the average time (95% confidence interval) of the TTUG test for the three age groups assessed in our study was: 8.1 (7.1–9.0) s for people aged 60 to 69, 9.2 (8.2–10.2) s for people aged 70 to 79, and 11.3 (10.0–12.7) s for people over 80 [28].

According to Wamser et al. (2015), there is a correlation between faster performance on the TTUG test and greater muscle strength, walking speed, and functional ability. On the other hand, a longer time to complete the test is associated with lower functional mobility, suggesting a greater likelihood of falls among individuals [16].

Older people tend to take longer to complete the TTUG test [15,16]. This phenomenon, which can be observed in the performance of the sample evaluated, can be attributed to physiological and biomechanical changes, such as a reduction in muscle mass, a greater accumulation of adipose tissue, a reduction in strength and a loss of flexibility which occur over the years, directly influencing the speed and agility of movements [16,17].

As such, the syndrome of frailty in the elderly is a physical vulnerability related to age, resulting from the deterioration of biological stability and the body’s ability to adapt to new stressful situations [1]. This increases the vulnerability to falls and the loss of functional independence [1]. However, most experts agree that frailty syndrome and sarcopenia can be reversed. They are linked to the performance of the musculoskeletal system, allowing rehabilitation through the restoration of physical capacity via regular physical exercise [4,5,19].

Additionally, regular and targeted interventions, such as muscle strengthening and balance activities, can significantly improve the quality of life for the elderly [1,19]. Studies show that exercise not only helps in recovering muscle mass but also improves bone density, flexibility, and cardiovascular endurance, all essential for maintaining functional independence and reducing the risks associated with frailty and sarcopenia [19].

Considering the specific context of elderly individuals participating in public physical activity programs, the results found may have been shaped by the very nature of the interventions carried out in the program, with factors such as muscle strength, flexibility, and balance preserved and acquired through regular physical exercise. However, the high anthropometric measurements indicate that, even with the benefits of physical activity, there are significant challenges to be faced in managing weight and abdominal fat among the elderly.

Additionally, it is important to consider that physical activity programs may need specific adjustments to effectively address abdominal obesity, which is a significant risk factor for cardiovascular and metabolic diseases [21]. The inclusion of nutritional strategies, personalized guidance, and training programs focused on reducing body fat can enhance the benefits already offered to the elderly, contributing to healthier and more active aging.

A possible limitation of our study is the small size of the male sample evaluated. This limitation is due to the smaller number of elderly males participating in the study’s physical activity project. Therefore, the smaller representation of males should be taken into account when interpreting the results, recognizing that the sample reflected the characteristics of this specific sub-population, which has a smaller number of male participants.

Another limitation pertains to the distribution of the participants’ age groups. After dividing the sample into three age groups, we observed that the ≥80 age group had a relatively low number of individuals. It is important to note that in studies involving older age groups, especially those ≥80 years, it is common to find a smaller number of participants. This is because the life expectancy of the world’s population, and, in particular, that of Brazilians, tends to be lower for octogenarians. In addition, this age group generally requires greater care, which makes it difficult for them to participate in public physical exercise programs like the group assessed in our study.

In addition, the cross-sectional design of the study is another limitation identified, as it did not allow us to follow the population evaluated over time, making it impossible to establish cause and effect relationships.

## 5. Conclusions

The results of this study indicate that, although many elderly people had anthropometric measurements above the appropriate averages, the correlations involving the TTUG test with BMI, WC, and WHtR were weak in this sample. This suggests that these results did not negatively influence TTUG performance, indicating that factors other than BMI and abdominal fat accumulation may have a greater influence on test performance.

However, it is important to acknowledge that the sample was limited in size and not fully representative, especially in the age group ≥80 years and among men. This limitation highlights the need for future studies with larger and more balanced samples to confirm these results and provide a more accurate view of the interactions between body composition, cardiovascular health and physical function in the elderly population.

## Figures and Tables

**Figure 1 ijerph-21-00993-f001:**
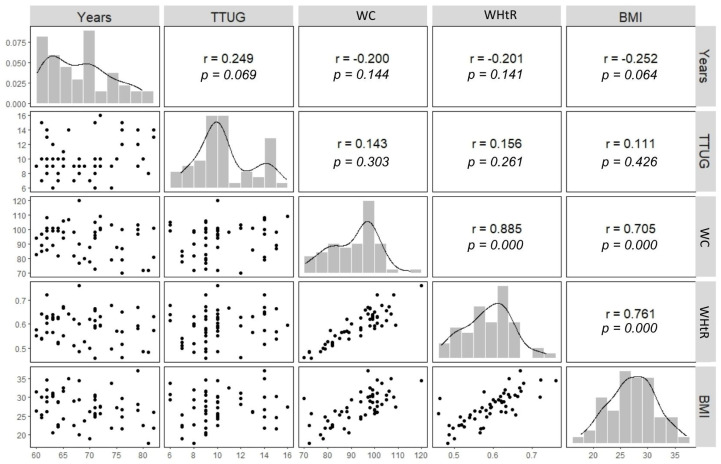
Statistical analysis and the correlations between the times used in the TTUG test and the four anthropometric variables of the sample. TTUG = Timed Up and Go test; BMI = body mass index; WC = waist circumference; WHtR = waist-to-height ratio. Source: the authors.

**Table 1 ijerph-21-00993-t001:** Anthropometric Characteristics and Results of the Timed Up and Go Test.

Variables	N 29 (52%)	N 22 (41%)	N 4 (7%)	Total N 55 (100%)
Age (Years)	60–69	70–79	≥80	
Weight (kg)	69.3 ± 11.1	68.1 ± 12.3	49.5 ± 10.3	67.2 ± 12.5
Height (cm)	156.9 ± 6.5	158.5 ± 10.4	149.5 ± 6.8	156.9 ± 8.5
BMI * (kg/cm)	28.1 ± 4.1	27.1 ± 3.9	21.1 ± 2.9	27.2 ± 4.2
WC * (cm)	94.6 ± 9.8	93.2 ± 11.1	81.5 ± 11.8	93.0 ± 11.0
WHtR * (cm/cm)	0.6 ± 0.1	0.5 ± 0.1	0.5 ± 0.1	0.6 ± 0.1
TTUG * (seg.)	9.7 ± 2.1	10.6 ± 3.1	11.2 ± 2.3	10.3 ± 2.5

* N = number; ± = standard deviation; BMI = body mass index; WC = waist circumference; WHtR = waist-to-height ratio; TTUG = Timed Up and Go test. Source: the authors.

## Data Availability

The data supporting the results reported in this study will be available from the corresponding author of the article. Please contact Adalgiza Mafra Moreno at adalgizamoreno@hotmail.com for access to the data.

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
