# Peer review of "Associations between Cardiovascular Risk Factors and Timed Up and Go Test for Elderly Participants in Public Physical Activity Programs"

_ijerph, 2024, doi:10.3390/ijerph21080993_

Round 1

Reviewer 1 Report

Comments and Suggestions for Authors

Comments and Suggestions for Authors

The authors developed an interesting study for the reader of International Journal of Environmental Research and Public Health. A study was developed, which aimed to investigate the associations between anthropometric indicators of cardiovascular risk and performance in the TUG test in elderly participants in public physical activity programs. However, the submitted article presents several scientific gaps and requires several improvements in the writing of the different chapters, particularly in terms of methodology and presentation of results. My overal recommendation is reconsider after major revisions. I leave some comments to help the authors improve the article.

Abstract:

- Line: 19-21. The writing seems unclear to me. To improve. For example: "correlations were found between the variable x and y...".

Introduction:

- Bibliographical references are missing in several parts of the introduction. Example: line 30, 34, 38, 43, 49, 57, 62, 64...

Materials and Methods:

- Line 83: how many participants were male and female? What is the average age overall and for each gender? add information.

- The inclusion and exclusion criteria are confusing. To improve.

- Was any type of procedure carried out to calculate the sample size? add information.

- What type of physical exercise do participants practice? frequency, duration, intensity...

- Line 112: "distance of 3 meters". Are you sure it's 3 meters?

- Add references to the tests and materials used.

- Line 102-113: separate the different instruments and tests by paragraphs and better describe the entire procedures used in each of the tests applied.

- Line 115-118: no descriptive statistical analysis was performed (means, standard deviation...)? add information.

- What computer software was used to carry out statistical analysis? add information.

- For better understanding, this chapter should be divided into sub-chapters (study design, participants, outcomes measurements, statistical analysis).

Results:

- Line 120: review formatting according to the journal standards.

- The tables presented are not formatted in accordance with the magazine's standards.

- The graph presented is not formatted in accordance with the magazine's standards.

- In the table, why was the sample divided by age group? Is it part of the study objective?

- The graph presented is of poor quality and uncomfortable for the reader.

- Line 121: wasn't the sample made up of 54 participants?

- The data is not described in a perceptible way. Details are missing. Example: strong correlations, positive correlations...

Discussion:

- The discussion must begin with a description of the objective of the study.

- The presentation of results must be in the "results" chapter. In the discussion, a brief summary of the most relevant results should be presented.

- Not all results found were discussed.

- The first paragraphs of the discussion do not make sense in this chapter. I think it would make more sense for them to be in the "introduction".

Conclusions:

- Line 260-261: this conclusion cannot be drawn from this study, because there was no type of intervention.

Reference:

- References are not formatted in accordance with the journal's standards.

- More recent references must be added. Less than 10 years.

Author Response

Main Points: Reviewer 1

Abstract:

  • Line: 19-21. The writing seems unclear to me. To improve. For example: "correlations were found between variables x and y...".

Response: We agree with the suggestion. We have revised the abstract to clarify the text.

Introduction:

  • Bibliographic references are missing in several parts of the introduction. Example: line 30, 34, 38, 43, 49, 57, 62, 64...

Response: References have been inserted in the revised manuscript.

Materials and Methods:

  • Line 83: How many participants were men and women? What is the general average age and for each gender? Add information.

Response: This information has been inserted starting from line 233.

The inclusion and exclusion criteria are confusing. To improve.

 Response: “Inclusion and exclusion criteria were as follows: Participants in the exercise program for more than 30 consecutive days, in the proposed age group, and meeting specific inclusion criteria. These criteria included having independent locomotion, preserved cognitive status, and the ability to understand and follow instructions related to performing the Timed Up and Go (TTUG) test.” The exclusion criteria involved elderly people with functional disabilities, such as the need to use a “such as those requiring” wheelchair or walking aids, individuals with severe visual or hearing difficulties, neurological sequelae and imbalances. Participants with flu-like symptoms were also excluded, in order to avoid the possibility of contagion among the other participants. In addition, flu-like symptoms such as fever, fatigue, muscle pain and breathing difficulties can compromise the safety and well-being of participants and significantly affect physical performance and the ability to perform the TTUG proposed in the study, negatively influencing the results of the assessments (23).The inclusion and exclusion criteria have been rewritten starting from line 151.

  • Was any procedure performed to calculate the sample size? Add information.

Response: The sample selection procedure has been better explained starting from line 145.

  • What kind of physical exercise do the participants practice? Frequency, duration, intensity...

 Response: The type of physical exercise that the participants practice has been explained from line 127 to line 141.

  • Line 112: "distance of 3 meters". Are you sure it's 3 meters?

Response: It has been better explained how the TTUG was performed and how the 3 meters of the test were marked starting from line 203.

  • Add references to the tests and materials used.

Response: These references have been inserted starting from line 198.

  • Lines 102-113: Separate the different instruments and tests by paragraphs and better describe all the procedures used in each of the applied tests.

Response: This information has been inserted starting from line 123.

  • Line 115-118: No descriptive statistical analysis was performed (means, standard deviation...)? Add information.

Response: This information has been inserted starting from line 227.

  • Which computer software was used to perform the statistical analysis? Add information.

Response: This information has been inserted starting from line 227.

  • For better understanding, this chapter should be divided into subchapters (study design, participants, outcome measurements, statistical analysis).

Response: These divisions have been made starting from line 121.

Results:

  • Line 120: Format the review according to the journal's standards.

Response: This has been done.

  • The tables presented are not formatted according to the journal's standards.

Response: They have been formatted.

  • The graph presented is not formatted according to the journal's standards.

Response: It has been redone.

  • In the table, why was the sample divided by age group? Is it part of the study objective?

Response: This information has been inserted starting from line 278.

  • The graph presented is of low quality and uncomfortable for the reader.

Response: It has been redone with better quality.

  • Line 121: Wasn't the sample composed of 54 participants?

Response: Typing error, it has been corrected to 55 participants.

  • The data are not described in a perceptible way. Details are missing. Example: strong correlations, positive correlations...

Response: This information has been inserted starting from line 267.

Discussion:

  • The discussion should start with a description of the study objective.

Response: This information has been inserted starting from line 261.

  • The presentation of the results should be in the "results" chapter. In the discussion, a brief summary of the most relevant results should be presented.

Response: This information has been inserted.

  • Not all the results found were discussed.

Response: This information has been inserted.

  • The first paragraphs of the discussion do not make sense in this chapter. I think it would make more sense if they were in the "introduction".

Response: This has been changed.

Conclusions:

  • Line 260-261: This conclusion cannot be drawn from this study as there was no intervention.

Response: This conclusion has been changed starting from line 368.

References:

  • The references are not formatted according to the journal's standards.

Response: They have been changed.

  • More recent references should be added. Less than 10 years.

Response: They have been added.

Reviewer 2 Report

Comments and Suggestions for Authors

The authors aimed to evaluate the relationship between anthropometric variables and the time taken to complete a TUG test in older volunteers . The topic is important given the known risk of falls in this age group. However, there are several concerns 

Major concerns

1. What the authors were trying to investigate seems already known. The authors need to provide a very clear and a strong justification as to why it was important to undertake the study. This can be added in the introduction section of the manuscript.

2. While the discussion tried to compare their results with previous studies, no effort was done trying to discuss biological plausibility of the results. The authors found strong correlation between TUG time and a number of anthropometric variables including waist circumference, but the biological plausibility of these relationships have not been sufficiently discussed. Particularly, plausibility between abdominal adiposity and falls, etc.

3. The statistical methods have been shallowly described in the methods section, and statistics incompletely presented in the results section. For instance the authors explained Pearson correlation has been used without stating whether all their results were normally distributed. Pearson correlation test is strictly for normally distributed data. No p values have been shown on the results section (including tables and figures) to show whether the relationships found were statistically significant.

Even though one can assume what is shown on the graphs are R values, but this has not been presented/shown in the figures or figure legend or anywhere in the results section. The presentation of the figures need to be improved for clarity.

Other comments

Unless if the journal requires such presentation, the results section of the ABSTRACT is very poorly described. In fact only values are presented and no description of results/findings has been given.

While the authors have shown the long form of WC as waist circumference on tables and figures, what is presented is CC. So one can only assume that is what is representing waist circumference. Additionally, the abbreviation Rcet is not defined anywhere, even though it appears on the table and figures.

There is a repitation of exclusion criteria on Line 89 of the material and methods 

Observational cross sectional and analytical study is not at all clear.

Why did the authors excluded those with flu like symptoms? 

Comments on the Quality of English Language

English language is OK, only trivial typos need to be rectified.

Author Response

  • Main Points: Reviewer 2
  • Major Concerns

  • What the authors were trying to investigate seems to be already known. The authors need to provide a very clear and strong justification for why it is important to conduct the study. This can be added to the introduction section of the manuscript.

Response: We appreciate the comment. We added more details, explaining the justifications.This has been added starting from line 88.

  • While the discussion tried to compare their results with previous studies, no effort was made to try to discuss the biological plausibility of the results. The authors found a strong correlation between TUG time and several anthropometric variables, including waist circumference, but the biological plausibility of these relationships was not sufficiently discussed. Particularly, the plausibility between abdominal adiposity and falls, etc.

Response: This information has been added starting from line 325.

  • The statistical methods were superficially described in the methods section, and the statistics were incompletely presented in the results section. For example, the authors explained that Pearson's correlation was used without stating whether all their results were normally distributed. Pearson's correlation test is strictly for normally distributed data. No p-values were shown in the results section (including tables and figures) to show whether the relationships found were statistically significant.

Response: This information has been added.

  • Although it can be assumed that what is shown in the graphs are R values, this was not presented/shown in the figures or figure legend or anywhere in the results section. The presentation of the figures needs to be improved for greater clarity.

Response: This information has been added.

  • Other Comments

  • Unless the journal requires such presentation, the results section of the ABSTRACT is very poorly described. In fact, only values are presented and no description of results/findings was given.

Response: This information has been added.

  • Although the authors have shown the long form of WC as waist circumference in tables and figures, what is presented is CC. So, it can only be assumed that this is what is representing the waist circumference. Additionally, the abbreviation Rcet is not defined anywhere, although it appears in the table and figures.

Response: This information has been added starting from line 197.

  • There is a repetition of exclusion criteria on Line 89 of the materials and methods.

 Response: This correction has been made.

  • The cross-sectional and analytical observational study is not clear at all.

Response: This correction has been made.

  • Why did the authors exclude those with flu-like symptoms?

Response: This information has been added starting from line 161.

Reviewer 3 Report

Comments and Suggestions for Authors

The manuscript aimed to evaluate whether TTUG correlates with anthropometric parameters associated with cardiovascular risk. The correlation was described although expected.

Major points:

- The manuscript should bring more tests that are not expected to make associations. 

- The novelty is not clear neither how this could improve healthcare providers or physicians.

- Most of the sujbects are women. This is a big bias in the study since men is often above weight.

- This is a specific population study and must be reflected in the title. Additionally, are there any characteristics of Niteroi's population that could affect the conclusions ?

- The anthropometric indicators used here are not exclusive of cardiovascular risk. They are (even though related to) more related to overweight/obesity. Cardiovascular risk requires additional parameters.

- Could social status influence the results ? What were the social status, education ? Authors should mention this information.

Author Response

Main Points: Reviewer 3

  • The manuscript should bring more tests that should not make associations.

Response: This information has been added.

  • The novelty is not clear nor how this could improve healthcare professionals or physicians.

Response: This information has been added starting from line 88.

  • Most subjects are women. This is a significant bias in the study as men are usually overweight.

Response: The reasons for the composition of this sample have been added starting from line 352.

  • This is a specific population study and should be reflected in the title. Also, is there any characteristic of the Niterói population that could affect the conclusions?

Response: Yes, We added more details about the population study. this information has been added in line 96.

  • The anthropometric indicators used here are not exclusive to cardiovascular risk. They are (although related to) more associated with overweight/obesity. Cardiovascular risk requires additional parameters.

Response: The changes can be found of the revised manuscript. his information has been better explained and changed.

  • Social status may influence the results. What was the social status, education? The authors should mention this information of he revised manuscript..

Response: This information has been added in line 96.
